# The Most Demanding Exercise in Different Training Tasks in Professional Female Futsal: A Mid-Season Study through Principal Component Analysis

**DOI:** 10.3390/healthcare10050838

**Published:** 2022-05-02

**Authors:** Markel Rico-González, Daniel Puche-Ortuño, Filipe Manuel Clemente, Rodrigo Aquino, José Pino-Ortega

**Affiliations:** 1Department of Didactics of Musical, Plastic and Corporal Expression, University of the Basque Country, UPV-EHU, 48940 Leioa, Spain; markel.rico@ehu.eus; 2Department of Physical Activity and Sport, Faculty of Sport Science, University of Murcia, 30100 Murcia, Spain; daniel.p.o@um.es (D.P.-O.); pepepinoortega@gmail.com (J.P.-O.); 3Escola Superior Desporto e Lazer, Instituto Politécnico de Viana do Castelo, Rua Escola Industrial e Comercial de Nun’Álvares, 4900-347 Viana do Castelo, Portugal; filipe.clemente5@gmail.com; 4Instituto de Telecomunicações, Delegação da Covilhã, 1049-001 Lisboa, Portugal; 5Research Center in Sports Performance, Recreation, Innovation and Technology (SPRINT), 4960-320 Melgaço, Portugal; 6LabSport, Post-Graduate Program in Physical Education, Centre of Physical Education and Sports, Federal University of Espírito Santo, Vitória 29075-910, Brazil; 7BIOVETMED & SPORTSCI Research Group, University of Murcia, 30100 Murcia, Spain

**Keywords:** indoor football, data mining, data reduction techniques, women, load monitoring

## Abstract

The contextual factors related to training tasks can play an important role in how a player performs and, subsequently, in how a player trains to face a competition. To date, there has been no study that has investigated the most demanding exercise in different training tasks in female futsal. Therefore, this study aimed to determine the most demanding efforts during different training tasks in a cohort study conducted in professional biological women futsal players using principal component analysis (PCA). A total of 14 elite women futsal players (age = 24.34 ± 4.51 years; height = 1.65 ± 0.60 m; body mass = 63.20 ± 5.65 kg) participated in this study. Seventy training sessions of an elite professional women’s team were registered over five months (pre-season and in-season). Different types of exercises were grouped into six clusters: preventive exercises; analytical situations; exercises in midcourt; exercises in ¾ of the court; exercises in full court; superiorities/inferiorities. Each exercise cluster was composed of 5–7 principal components (PCs), considering from 1 to 5 main variables forming each, explaining from 65 to 75% of the physical total variance. A total of 13–19 sub-variables explained the players’ efforts in each training task group. The first PCs to explain the total variance of training load were as follows: preventive exercises (accelerations; ~31%); analytical situations (impacts; ~23%); exercises in midcourt (high-intensity efforts; ~28%); exercises in ¾ of the court (~27%) and superiorities/inferiorities (~26%) (aerobic/anaerobic components); exercises in full court (anaerobic efforts; ~24%). The PCs extracted from each exercise cluster provide evidence that may assist researchers and coaches during training load monitoring. The descriptive values of the training load support a scientific base to assist coaches in the planning of training schedules.

## 1. Introduction

Futsal (also known as indoor football) is an intermittent exercise, in which repeated efforts occur interspaced by periods of short recovery [1,2]. High-demanding efforts are associated with futsal, considering that, in this game, a lot of quick transitions of possession of the ball occur with implications for physical demands [3]. In the specific case of women’s soccer, total distance can vary between 1659 and 1547 m per half, while 41–42% of the distance is covered between 6.1 and 12 km/h, 6–7% between 15.5 and 18.3 km/h and 4–5% above 18 km/h [4]. The maximal sprint speed in women’s futsal is between 26 and 31 km/h, depending on the half [4]. Therefore, elite women futsal players usually possess high levels of maximal oxygen uptake (~58.7 mL/kg/min) [4] and sprinting abilities, since they can achieve times of 1.6s, 2.98s and 4.16s in 10, 20 and 30 m sprint tests [5].

Therefore, understanding the training context is important for characterizing regular stimuli on futsal players and identifying how they are managed for achieving specific goals [6]. Although there is a possible non-linear effect between load and individual responses to stimuli in futsal [7], it is expected that specific determinants of intensity can explain some performance and biological variations [8,9]. As an example, a study conducted on women soccer players examined the changes in salivary immunoglobin A, cortisol and upper respiratory tract infection and tested the relationship with training intensity [10]. In this study [10], it was found that a higher training intensity had significant relationships with an increase in upper respiratory tract infection, thus suggesting the impact of the training process on acute physiological and immunological responses. Similarly, a study conducted in women’s soccer revealed that moderate-intensity training enhanced cytokine patterns [11]. 

Naturally, the contextual factors associated with the futsal competitive schedule [12], moment of the season [13] and match demands will modulate the training process. Thus, it is expected that different training drills occurring in training scenarios may result in different outcomes, and the determinants of exercise can be different based on the futsal drill typology [6,14,15]. The variation in those determinants should be understood considering the dose–response effect. Despite the importance of considering the dose–response effect of specific intervention programs on changes occurring in fitness variables in futsal players [16,17], it is also important to understand the characteristics of the training intensity distribution in the microcycle [18] and the individual effects of specific training exercises on the players [19]. Although there has been some research describing the variations in intensity within and between microcycles [18,19], it did not consider the specific impact of types of exercises and the contribution of types of exercises for the intensity imposed on the players. Although not found in women’s futsal, this characterization of the intensity imposed by different types of exercises was described in soccer, suggesting that the highest volume of training and the greatest locomotor demands (physical demands) imposed on the players were based on game-based drills [20].

Descriptive analysis is important to understand how the intensity is distributed across the different training drill types. However, relationships between drills and locomotor demands may improve the capacity to understand the training process’s determinant outcomes. Considering the great number of variables typically extracted by microelectromechanical systems, it is important to find a reduction technique that allows identifying the most important variables to be analyzed. One of the approaches to do that is applying principal component analysis (PCA) [21] that, in this context, may play a role as a data reduction technique to highlight the most demanding efforts. In male futsal players, PCA allowed identifying that using 3–4 principal components was the most appropriate approach to determine players’ performance in matches [22]. In the case of women’s futsal, PCA allowed identifying 22 external and internal intensity measures that may be used for intensity monitoring during official matches [23]. Despite these findings regarding PCA in official matches, there is no related study (as far as we know) that has applied PCA in training drills and sessions of women futsal players. This may help to understand which measures are important to detect for monitoring training intensity in women’s futsal training sessions. Therefore, the purpose of this study was to determine the most demanding efforts during different training tasks in a cohort study conducted in professional female futsal players considering a range of local positioning system (LPS) (e.g., distances covered at different velocities, accelerations and decelerations) and heart rate-based variables.

## 2. Materials and Methods

### 2.1. Design

The data were registered from daily player monitoring in which the athletes’ motion was routinely measured throughout the season. Measures were gathered by a real-time motion tracking system which includes a local positioning system (LPS) device, based on UWB technology, and an inertial measurement unit (IMU; WIMU PROTM, RealTrack Systems, Almeria, Spain). In addition, one GARMIN HRM-PRO chest strap (Garmin Ltd., Olathe, KS, USA) was located below the chest. Data were sent to the WIMU PRO device, which stores all data sets. WIMI PRO devices were daily attached to the players’ upper back in a pocket attached to a tight-fitting garment, placed between the scapulae at the T2–T4 levels to avoid unwanted movements. The tight-fitting garment was the same for each player in each game. The same routine was implemented each day, and both protocol and task control were daily checked by the same team staff. 

In this cohort study, all tasks considered throughout the season during 70 training sessions (from September to February) were clustered into different groups depending on their characteristics (see Table 1). During these training sessions, an LPS and microelectromechanical system (MEMS) were used to extract information from more than 250 variables, which may be summarized as accelerations/decelerations, short-time efforts, distance covered at different intensities and physiological variables such as heart rate. Through a multivariate reduction technique (PCA), the more relevant variables were extracted, highlighting the most demanding efforts by professional female futsal players in each group of tasks.

### 2.2. Participants

Data were collected from 14 professional biological women futsal players (means ± standard deviations: age = 24.34 ± 4.51; height = 1.65 ± 0.60 m; body mass = 63.20 ± 5.65 kg) from a Spanish 1st division top-level team during the 2020–2021 pre-season and in-season periods (from September to February). This sample was considered due to the high number of individual observations and the real-world scientific practice in high-level settings. Players were allowed to participate in a training session if they were healthy and they did not have injuries or did not participate in a post-injury readaptation program. Due to the critical situation as a result of severe acute respiratory syndrome (SARS-CoV-2), players were excluded from the analysis when they were confined. All players were notified of the aim of the study and procedures in accordance with the Declaration of Helsinki. The study was approved by the ethical committee of the University of Murcia (protocol code 3180/2020).

### 2.3. Data Collection

To conduct a strict description of the use of technology, a recently published survey was followed [24]. For the use of an ultra-wideband (UWB)-based LPS, 21 points out of 23 were explained, while for the use of a MEMS, 17 points out of 20 were detailed. The rest of the items cannot be explained because the information was not available to the authors.

### 2.4. Measures

The UWB technology operates on a much wider frequency band than other traditional radio communication technologies (at least 0.5 GHz), and a previous study did not report any problems in UWB-based tracking system accuracy in multipath conditions (i.e., 28 devices turned on) (Bastida Castillo et al., 2018). This system holds both FIFA International Match Standard and Quality certificates [25]. Each device consists of an internal microprocessor, 2 GB flash memory and a high-speed USB interface, to record, store and upload data.

#### 2.4.1. Ultra-Wideband

From the UWB distance covered at different intensities, velocity and load indicators were registered (see Table 2). The data were recorded in a training space far from metallic materials. The UWB system was composed of a reference system and tracked devices carried by the players. The antennae are transmitters and receivers of radio frequency signals. The antennae (mainly the master antenna) computerize the position of the devices that are in the play area, while the device receives that calculation using time difference of arrival (TDOA). The eight antennae were installed five hours before the match, forming an octagon for better signal emission (4.5 m from the perimeter line for antennae located in the corners, and 5.5 m from the perimeter line for the antennae located in the middle of the court and behind the goals) and reception at a height of 3 m, and held by a tripod. Once installed, they were switched on one by one, with the master antenna turned on last. From that moment, it was necessary to respect a 5-min protocol to avoid technology lock [24]. To allow data time synchronization, the master antenna managed the time using a common clock which allows data recording at the same time. When all devices were switched on in the center of the reference system, a process of automatic recognition between antennae and devices was carried out for 1 min. In this study, the raw data were recorded at an 18 Hz sampling frequency because low frequencies have been shown to have a lower quality of measurement, and 18 Hz with UWB has not shown less accuracy because of noise problems. The conditions were maintained with low temperatures, humidity gradients and slow air circulation to allow easier positioning. This UWB system has demonstrated valid and reliable measures during continuous situations [26].

#### 2.4.2. Inertial Measurement Units

From the IMU, acceleration, deceleration, impact and landing-related variables were registered (see Table 2). The validity of the inertial measurement unit (IMU; WIMU PROTM, Real Track Systems, Almeria, Spain) was assessed in a previous study [27]. The results of the validity in this study were satisfactory and had a bias of 0.0006 ± 0.0018 s. The calculation of the velocity was conducted through differential Doppler and the acceleration was calculated from velocity. Finally, the minimum effort duration and minimum speed for avoiding unrealistic data mentioned were defined by the manufacturer to avoid outliers.

### 2.5. Performed Training Task Clusters and Variables

Positional data recorded during training tasks (Table 1) extracted from PCA are summarized in Table 2. 

The training task clustering followed a technical classification proposed by team staff, which focused on the rationale of the classification. Further, the task clusters were established depending on the efforts in each of them. In cluster number one, warm-up exercises were classified, which players performed at the beginning of each training session. In cluster number two, the harbor exercises were established (i.e., low physical stress), where team staff halted the exercise to explain tactical information. Clusters three, four and five focused the classification’s rationale on the number of players involved (individual training tasks, pure opposition, opposition and collaboration and opposition and collaboration with numerical unbalances). Additionally, tasks with opposition and collaboration (from 3 vs. 3 to 6 vs. 6) were classified into 3 clusters depending on space (i.e., midcourt, ¾ of the full court and full court). These clusters included the most tactical and technical training enrichment exercises for the futsal players, close to the real situations. Finally, cluster number six refers to all these exercises, usually with a high-stress situation, where players with superiority/inferiority need to make decisions with high fatigue and risk. 

The training exercises non-characterized by task constraints were excluded from the analysis. In addition, abbreviations and definitions from the variables extracted from registered training tasks and that formed PCs are described in Table 2. See Figure 1 for details regarding the experimental protocol. 

### 2.6. Statistical Analysis

To describe the protocol of PCA, the survey purposed by Rojas-Valverde et al. [28] was followed. The protocol was explained step by step in Oliva-Lozano et al. [29], which contains the following steps: field data collection, software outcome (inclusion of all variables measured), correlation matrix exploration (all variables correlated between each other), assumption confirmation (variables were centered and scaled (Z-Score), suitability confirmed by KMO and Bartlett sphericity values), PCA (eigenvalues greater than 1 were included for extraction), loading groupings in each PC (loadings > 0.7 were considered for PCA grouping and inclusion) and final PCA outcome (each PCA was presented and grouped by loading) [29]. From more than 250 variables recorded, 12 (cluster 1), 32 (cluster 2), 32 (cluster 3), 28 (cluster 4), 27 (cluster 5) and 31 (cluster 6) variables were explored using a correlation matrix in order to select the most representative variables, and those with correlations r < 0.7 between variables were considered for extraction [30]. After exclusion of variables with variance = 0, 22–21 variables were scaled and centered using Z-Scores. PCA suitability was confirmed through the Kaiser–Meyer–Olkin value (KMO = 0.64–0.78) and Bartlett sphericity test significance (*p* < 0.05) [31]. Eigenvalues > 1 were considered for the extraction for each principal component, and a varimax orthogonal rotation method was used to identify high correlations between components to offer different information. PC loadings > 0.6 were considered for extraction, and when a cross-loading was found between PCs, only the highest factor loading was retained [31].

## 3. Results

From 250 variables extracted from the tracking system and MEMS, 8 main variables and load indicators were extracted, which contain a total of 42 sub-variables: (1) distance covered at different intensities (Dist. (m·min^−1^), Expl dist and HSR Abs (m/min)); (2) heart rate-related variables (maximum HR, relative HR and time spent at heart rates of different intensities); (3) velocity at different intensities (from 0 to 6, from 18 to 21, from 21 to 24 and maximum velocity); (4) variables related to accelerations (Acc/min, distance accelerating, maximum acceleration and absolute accelerations at different intensities (five levels)); (5) decelerating-related variables (six levels of decelerations at different intensities); (6) impacts (measured in both G (9.8 m·s^2^) and number per minute (four levels)); (7) a variable related to landing (from 5 to 8 landings per minute); and (8) six load indicators (player load, power metabolic, HML (two levels), DSL, HBD and HIBD). However, the contributions of these sub-variables to explaining players’ external and internal load responses differed between types of exercises (Table 3). Descriptive statistics (e.g., mean, median, standard deviation, percentile) of each variable extracted from PCA for each exercise cluster are described in Appendix A.

The preventive exercise cluster (exercise cluster 1) was composed of 6 PCs formed by 14 variables, which explained a total of 70% of players’ behavior during these exercises. In this case, the first PC was composed of five sub-variables related to accelerations: MAX Acc (m·s^2^), Acc Abs (2–3)/min, Acc Abs (4–5)/min, Acc Abs (5–6)/min and Dec Abs (−6, −5)/min. In addition, cluster 2 was composed of 6 PCs formed by 14 variables, which explained a total of 68% of players’ physical behavior during these exercises. In this case, the first PC was composed of two variables related to impacts: Impacts (0–3) and Impacts (3–5) G. 

Cluster 3 (exercises performed in 20 × 20 meters) was composed of 7 PCs formed by 19 variables, which explained a total of 76% of players’ physical behavior. In this case, the first PC was composed of four variables related to high-intensity efforts: Dist Expl, HR % (90–95), HIBD and Acc Abs (3–4)/min. The efforts performed in 20 × 28 meters (exercise cluster 4) were explained by 15 sub-variables included in 7 PCs, which explained a total of 73% of players’ physical performance during these tasks. In this case, the first PC was composed of five sub-variables about aerobic and anaerobic components: Dist Expl, HR % (50–60), HR % (80–90), Player Load /min and Power Metabolic.

Cluster 5 (exercises in full court) was composed of 5 PCs formed by 13 variables, which explained a total of 66% of players’ physical behavior during exercises performed in 20 × 40 m. The first PC was composed of two sub-variables related to anaerobic efforts: Dist Expl and HBD. Finally, during the exercises based on superiorities/inferiorities (cluster 6), players seemed to perform Expl dist, Dist (m·min^-1^), HR % (50–60), MAX HR (bpm) and Rel HR %. These five sub-variables were clustered into seven PCs that explained 75% of the variance.

## 4. Discussion

This is the first study to determine the most demanding efforts, based on external and internal loads, during different training tasks in professional female futsal players. The main findings were as follows: (i) a combination of 5–7 PCs from a total of 1–5 levels of main variables were required to explain 65–75% of the variance of training load during different tasks; (ii) a total of 13–19 sub-variables explained the players’ efforts in each training task group; (iii) the first PC (~31% of variance) of the preventive exercises (cluster 1) was composed of sub-variables related to accelerations; (iv) the best PC (~23%) to explain the variance of players’ load responses during analytical situations (cluster 2) was composed of impacts (0–5 G); (v) the load responses during exercises in the midcourt (cluster 3) were explained (~28%) mainly by sub-variables related to high-intensity efforts; (vi) the variances of efforts performed in 20 × 28 meters (cluster 4) and superiorities/inferiorities (cluster 6) were better explained (~27% and ~26%, respectively) by aerobic and anaerobic components; (vii) the first PC (~24% of the variance) of the exercises in full court (cluster 5) included sub-variables related to anaerobic efforts. These results provide new information for researchers and practitioners on the load responses and most demanding efforts during different training tasks, assisting during the planning of load prescriptions in female futsal.

A previous study on amateur male futsal players (third Spanish division) showed that a combination of 3–4 PCs from a total of 8–10 external load variables (e.g., total distance in m/min; high-speed running in m/min; explosive distance; impacts 8–100 G; jumps; acceleration/deceleration) were required to explain 63.5% of the match physical variance [22]. In young elite basketball players, three PCs and six variables were extracted (e.g., average/maximum acceleration, landing 8–100 G, relative distance, jumps) [32]. The basketball and futsal sports similarities (e.g., game space and invasion team sports) can explain some coincident variables extracted from PCA (e.g., acceleration, jumps). 

In Spanish professional soccer players, ~66% of the total variance of the match external load was explained by 11 variables, such as total distance covered at different speeds (including low/high-speed running) and accelerations/decelerations [29]. In soccer training (e.g., small/medium/large-sided games) and official match contexts, three PCs summarized several external load variables, where the first PC explained 39–44% of the total variance using five variables: distance covered in m/min; the number of accelerations/decelerations at high intensity; average metabolic power (highest weight); high metabolic load distance [33]. Therefore, these results suggest that internal logic constraints (e.g., game space and goal location) influence the variables extracted from PCA in training and match contexts in team sports. However, the previously cited studies included male participants. Therefore, direct comparisons between training load in male and female players should be carried out with caution.

A previous study showed that elite female professional futsal players covered ~3.200 m (~12% in high-speed running) during a one-off game [4]. Analysis from 1 month of training in amateur female futsal showed that players reached ~75% (±11) of the HRMAX and ~15% for time spent at ≥ 90% of the HRMAX during training sessions [34]. In addition, small-sided games in the midcourt (exercises included in cluster 3) resulted in high values of internal load (e.g., ~81% HRMAX) in male semi-professional futsal players from a Portuguese first division team [35]. To date, to the best of our knowledge, no previous studies have shown both external and internal load responses during different training tasks in elite professional female futsal. Here, the most demanding efforts were fully described in six exercise clusters, demonstrating the key physical performance indicators in a plethora of exercises. Therefore, researchers and coaches could select the variables to monitor physical performance according to the sub-variables from PCs presented in each exercise cluster (see Table 3).

Furthermore, the current study offers evidence-based approaches to physical conditioning training in elite futsal. The descriptive values described in Appendix A show the set of external and internal load responses in each exercise cluster. Exercises performed in game-based approaches (cluster 3: midcourt; cluster 4: ¾ of the court; cluster 5: full court) presented the highest distance covered in explosive actions (i.e., Explosive Dist; median ~9–11 m·min^−1^). These values (~12 m·min^−1^) are similar to those verified during official futsal matches played by male amateur players [22]. This study also verified that small-sided games (cluster 3; 20 × 20 m) demonstrated the lowest values of impacts 5–8/min (~3.5) compared to medium- (cluster 4; 20 × 28 m; ~4.1) and large-sided games (cluster 5; 20 × 40 m; ~4.5). In soccer players, smaller small-sided game formats mainly increased acceleration/deceleration and internal load measures [36]. In addition, small relative areas (80 m^2^ per player in futsal matches) are known to under-stimulate the distance covered at high speed [37]. Here, exercises in the midcourt (20 × 20 m) and ¾ of the court (20 × 28 m) presented similar values of time spent (24–26%) from 80–90% of HRMAX. These results are in line with a previous study [14] on top-level male futsal players from a first division Brazilian team (36 × 20 m court: 74.1 ± 3.6% of HRMAX; 31 × 19 m court: 73.7 ± 3.9% of HRMAX; and 25 × 15 m court: 70.5 ± 1.5% of HRMAX). This information could provide a scientific-based approach to assist coaches in planning training schedules.

## 5. Limitations of the Study

This study has some limitations. The major limitation was that the training load analysis was performed without considering the biological and physiological characteristics of the participants. In addition, the design was conducted only in the mid-season. Moreover, since some contextual factors such as exercises in different moments of a season (pre- vs. in-season; competitive schedule) [12,13], exact time of each training task or participant’s menstrual cycle may affect extracted outcomes, these results should be interpreted with caution. 

Further studies should investigate the most demanding efforts in multiple teams and full seasons, including the biological and physiological characteristics of the participants. However, this study advances several aspects from the previous literature about this topic. First, this is the first study to investigate both external and internal load responses during a myriad of training exercises in elite female futsal players. Second, the large sample of dependent variables (i.e., 250) enabled a deep understanding of physical demands during futsal training sessions. Third, the statistical approach (i.e., PCA) used allows researchers and practitioners to identify key physical performance indicators during training settings [28].

## 6. Conclusions

In summary, a combination of 5–7 PCs from a total of 1–5 levels of main variables were required to explain 65–75% of the variance of training load during different tasks. A total of 13–19 sub-variables explained the players’ efforts in each training task group. However, these variables differed between types of exercises. The first PCs to explain the total variance of training load during each exercise cluster were as follows: preventive exercises (accelerations; ~31%); analytical situations (impacts; ~23%); exercises in midcourt (high-intensity efforts; ~28%); exercises in ¾ of the court (~27%) and superiorities/inferiorities (~26%) (aerobic and anaerobic components); exercises in full court (anaerobic efforts; ~24%). Finally, the explosive actions performed during the game-based exercises grouped in clusters 3, 4 and 5 presented similar values to those previously observed during official futsal matches. Therefore, these tasks could be useful to represent match load during training settings.

## Figures and Tables

**Figure 1 healthcare-10-00838-f001:**
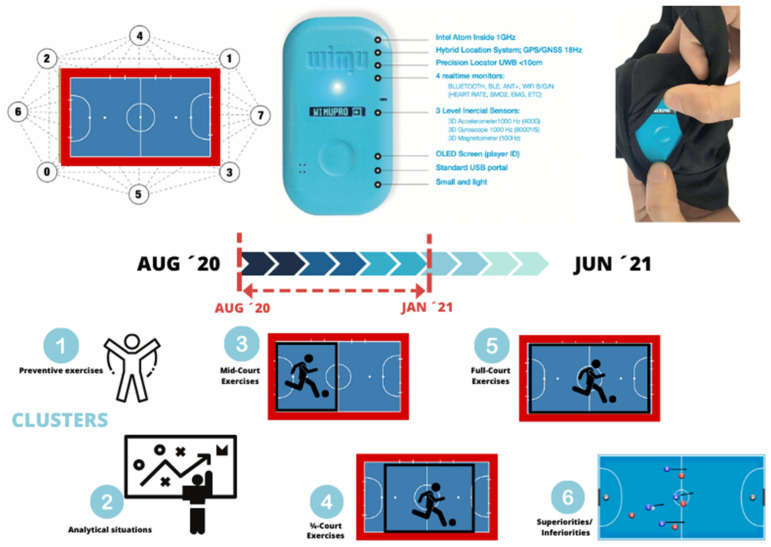
The experimental protocol and method.

**Table 1 healthcare-10-00838-t001:** Training tasks grouped into exercise clusters.

Cluster	Characteristics	Task Description
1	Preventive exercises (individual)	- Plyometric - Strength circuits - Tabata - Proprioception - Eccentric - Reactive velocity - Velocity/accelerative
2	Analytical situations	- Analytical tactical movements with and without opponents - Corners (without opposition) - Throw-ins - Shot on goal from fouls - Shot on goal from the penalty spot
3	Exercises in midcourt (20 × 20 m)	- Games (with and without wildcard) from 3 vs. 3 to 6 vs. 6 - Corner strategy (exercise with opposition starting from a corner)
4	Exercises in ¾ of the court (20 × 28 m)	- Games (with and without wildcard) from 3 vs. 3 to 6 vs. 6 - Counterattack exercises
5	Exercises in full court (20 × 40 m)	- Games (with and without wildcard) from 3 vs. 3 to 6 vs. 6 - Counterattack exercises
6	Superiorities/inferiorities	- Situations with numerical superiorities/inferiorities

**Table 2 healthcare-10-00838-t002:** External and internal training load variables.

		Abbreviation	Sub-Variables
1	Distance covered at different intensities	Dist (m·min^−1^)	Distance covered
		Expl dist	Distance covered at explosive intensity
		HSR Abs (m·min^−1^)	High-speed running
		HIBD (m·min^−1^)	High-intensity break distance with DEC > 2 m·s^−2^
2	Heart rate-related variables	HR % (50–60)	Time spent from 50 to 60 % of maximum heart rate
		HR % (70–80)	Time spent from 70 to 80 % of maximum heart rate
		HR % (80–90)	Time spent from 80 to 90 % of maximum heart rate
		HR % (90–95)	Time spent from 90 to 95 % of maximum heart rate
		HR % (>95)	Time spent up to 95 % of maximum heart rate
		MAX HR (bpm)	Maximum heart rate achieved
		Rel HR %	% of heart rate scored per minute
3	Velocities	Vel Abs (0–6) (m·min^−1^)	Absolute distance covered from 0 to 6 m·min^−1^
		Vel Abs (18–21) (m·min^−1^)	Absolute distance covered from 18 to 21 m·min^−1^
		Vel Abs (21–24) (m·min^−1^)	Absolute distance covered from 21 to 24 m·min^−1^
		Vel Max	Maximum velocity achieved by a player
4	Accelerations	Acc/min	Number of accelerations per minute
		Dist Acc	Distance accelerating
		MAX Acc (m·s^2^)	Maximum acceleration
		Acc Abs (0–1)/min	Absolute accelerations lower than 1 per minute
		Acc Abs (1–2)/min	Absolute accelerations from 1 to 2 per minute
		Acc Abs (3–4)/min	Absolute accelerations from 3 to 4 per minute
		Acc Abs (4–5)/min	Absolute accelerations from 4 to 5 per minute
		Acc Abs (5–6)/min	Absolute accelerations from 5 to 6 per minute
		Acc Abs (6–10)/min	Absolute accelerations from 6 to 10 per minute
5	Decelerations	Dec Abs (−1, 0)/min	Absolute decelerations lower than 1 per minute
		Dec Abs (−2, −1)/min	Absolute decelerations from 1 to 2 per minute
		Dec Abs (−4, −3)/min	Absolute decelerations from 3 to 4 per minute
		Dec Abs (−5, −4)/min	Absolute decelerations from 4 to 5 per minute
		Dec Abs (−6, −5)/min	Absolute decelerations from 5 to 6 per minute
		Dec Abs (−10, −6)/min	Absolute decelerations from 6 to 10 per minute
6	Impacts	Impacts (0–3) G	Impacts at intensity lower than 3 G
		Impacts (3–5) G	Absolute decelerations from 3 to 5 G
		Impacts (0–3)/min	From 0 to 3 impacts per minute
		Impacts (3–5)/min	From 3 to 5 impacts per minute
		Impacts (5–8)/min	From 5 to 8 impacts per minute
7	Landings	Landing (5–8)/min	From 5 to 8 landings per minute
8	Load indicators	Player Load /min	Player load extracted from accelerometer’s 3 axes
		Power Metabolic	Energy consumed kg·s
		HML (10–25.5) (m)	Distance covered from 10 to 25.5 m at 25.5 W/kg, which corresponds to 5.5 m/s^2^ or significant acceleration/deceleration efforts
		HML (25.5–35) (m)	Distance covered from 25.5 to 35 m at 25.5 W/kg, which corresponds to 5.5 m/s^2^ or significant acceleration/deceleration efforts
		DSL/min	Dynamic strength load. Total impacts with high intensity of 2 G.
		HBD (m·min^−1^)	

Note: G = 9.8 m/s^2^.

**Table 3 healthcare-10-00838-t003:** Representation of principal component analysis from external and internal load responses in different training tasks performed by elite professional female futsal players during mid-season.

PC	1	2	3	4	5	6	7
Cluster 1 (preventive exercises)							
Eigenvalue		11.5	9.8	7.0	5.7	5.2	
% Variance	31.4	42.9	52.8	59.9	65.6	70.7	
HR % (50–60)			0.798				
HR % (70–80)			0.771				
HR % (90–95)					0.792		
MAX Acc (m/s^2^)	0.766						
Acc Abs (2–3)/min	0.780						
Acc Abs (4–5)/min	0.798						
Acc Abs (5–6)/min	0.759						
Dec Abs (−6, −5)/min	0.716						
Impacts (0–3) G						0.850	
Impacts (0–3) min							
Impacts (3–5)/min				0.874			
Impacts (5–8)/min		0.917					
Impacts (8–100)/min		0.950					
Landing (5–8)/min		0.729					
Cluster 2 (analytical situations)							
Eigenvalue		13.8	10.5	8.2	6.9	5.3	
% Variance	23.4	37.3	47.7	55.9	62.8	68.1	
HR % (50–60)				0.736			
HR % (80–90)				0.817			
DSL/min					0.766		
Vel Abs (0–6) (m·min^−1^)		0.705					
Acc/min		0.903					
Acc Abs (0–1)/min		0.903					
Acc Abs (1–2)/min		0.737					
Acc Abs (2–3)/min			0.766				
Acc Abs (4–5)/min			0.799				
Dec Abs (−5, −4)/min			0.801				
Impacts (0–3) G	0.926						
Impacts (3–5) G	0.734						
Impacts (0–3) min						00.752	
Impacts (5–8)/min						0.855	
Cluster 3 (exercises in midcourt)							
Eigenvalue		14.9	9.4	7.3	6.4	4.9	4.2
% Variance	28.4	43.3	52.8	60.1	66.5	71.4	75.6
Expl dist (m)	0.867						
HR % (50–60)				0.736			
HR % (80–90)				0.720			
HR % (90–95)	0.802						
HR % (>95)							0.880
HIBD (m·min^−1^)	0.859						
Vel Abs (18–21) (m·min^−1^)			0.803				
Acc/min				0.900			
Dist Acc							.726
MAX Acc (m/s^2^)		0.730					
Acc Abs (3–4)/min	0.708						
Acc Abs (5–6)/min		0.738					
Acc Abs (6–10)/min		0.840					
Dec Abs (−1, 0)/min							
Dec Abs (−2, −1)/min					0.919		
Impacts (0–3) G						0.956	
Impacts (3–5) G						0.949	
Impacts (0–3) min			0.775				
Impacts (5–8)/min			0.837				
Cluster 4 (exercises in ¾ of the court)							
Eigenvalue		14.0	10.8	7.1	5.3	4.6	4.0
% Variance	26.9	40.8	51.6	58.7	64.0	68.6	72.6
Expl dist (m)	0.725						
HSR Abs (m·min^−1^)			0.807				
HR % (50–60)	0.713						
HR % (70–80)						0.842	
HR % (80–90)	0.764						
Acc/min					0.885		
Acc Abs (2–3)/min		0.700					
Acc Abs (3–4)/min		0.882					
Acc Abs (4–5)/min		0.760					
Acc Abs (6–10)/min				0.852			
Dec Abs (−1, 0)/min					0.854		
Impacts (3–5) G							0.802
Impacts (5–8)/min			0.928				
Player Load /min	0.816						
Power Metabolic (kg·s)	0.779						
Cluster 5 (exercises in full court)							
Eigenvalue		16.3	10.9	7.7	6.2		
% Variance	24.5	40.8	51.8	59.5	65.8		
Expl dist (m)	0.920						
HML (10–25.5) (m)			0.911				
HML (25.5–35) (m			0.861				
HBD (m/min)	0.854						
Vel Abs (18–21) (m·min^−1^)					0.748		
Acc/min				0.839			
Acc Abs (5–6)/min		0.714					
Acc Abs (6–10)/min		0.854					
Dec Abs (−2, −1)/min				0.909			
Dec Abs (−6, −5)/min		0.783					
Dec Abs (−10, −6)/min		0.704					
Impacts (0–3) G			0.880				
Impacts (5–8)/min					0.944		
Cluster 6 (superiorities/inferiorities)							
Eigenvalue		14.0	10.7	8.9	5.7	5.1	4.2
% Variance	26.4	40.4	51.1	60.0	65.7	70.9	75.1
Expl dist (m)	0.700						
Dist (m·min^−1^)	0.778						
HR % (50–60)	0.745						
HR % (70–80)						0.765	
MAX HR (bpm)	0.795						
Rel HR %	0.885						
Vel Abs (18–21) (m·min^−1^)			0.867				
Acc/min				0.781			
Dist Acc							0.727
MAX Acc (m·s^2^)					0.727		
Acc Abs (0–1)/min				0.852			
Acc Abs (2–3)/min		0.799					
Acc Abs (3–4)/min		0.864					
Acc Abs (4–5)/min		0.763					
Dec Abs (−4, −3)/min		0.752					
Dec Abs (−5, −4)/min		0.743					
Impacts (0–3) min			0.787				

## Data Availability

The data can be obtained from the corresponding author.

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
