# Peer review of "The Most Demanding Exercise in Different Training Tasks in Professional Female Futsal: A Mid-Season Study through Principal Component Analysis"

_healthcare, 2022, doi:10.3390/healthcare10050838_

Round 1

Reviewer 1 Report

General Comments

The paper entitled “The most demanding efforts in different training tasks in professional female futsal: A mid-season study through principal 3 component analysis” is relevant for picturing the training monitoring on woman football players. However, improvements should be considered, as the manuscript rational and readability (e.g. paragraphs quite long), details methods description and improvements on discussion.

Title

Is the word effort is the same than intensity or exercise? Here are a crucial aspects regarding the concept. Please, rewrite.

Abstract

A background regarding the rational of the study should be included.

Insert the number of participants and the main characteristics of them

Introduction

The introduction is quite long and some aspects should be reviewed. For example in the 3rd paragraph, none aspect regarding the football game is considered on training plain. In any momento of the introduction authors retract this important situation that is included in football players, the effort during the games. We know that the training of this athletes is based on the calendar of the games. Please, rewrite some parts of the introduction considering the above mentioned aspect.

In the last paragraph, authors should describe the principal variables that were included in PCA analysis in the others studies and justify the use of them.

Methods

Authors should explain how was the PCA method? Which variables were included and excluded? Authors should explain step by step.

Participants

Authors should explain how was the control of the games during the evaluation period.

Protocol

Authors should better explain the experimental protocol, focus on the efforts in different training tasks. A lot of information was described about the instruments, but a few information about the tasks proposed to evaluate the efforts.

A figure regarding the experimental protocol and the methods used should be included.

Measurements 

Authors should include the error of measurements using IMU.

Discussion

See the comments in the introduction section to improve the discussion.

Author Response

The paper entitled “The most demanding efforts in different training tasks in professional female futsal: A mid-season study through principal 3 component analysis” is relevant for picturing the training monitoring on woman football players. However, improvements should be considered, as the manuscript rational and readability (e.g. paragraphs quite long), details methods description and improvements on discussion.

RESPONSE: THANK YOU VERY MUCH FOR YOUR ACKNOWLEDGEMENT AND SUPPORTING OUR WORK. ALL YOUR RECOMMENDATION HAVE BEEN CONSIDERED.

Title

Is the word effort is the same than intensity or exercise? Here are a crucial aspects regarding the concept. Please, rewrite.

 RESPONSE: THANK YOU. YOUR SUGGESTION HAS BEEN FOLLOWED. “EFFORTS” HAS BEEN REPLACED BY “EXERCISE”.

Abstract

A background regarding the rational of the study should be included.

RESPONSE: YOUR SUGGESTION HAS BEEN FOLLOWED.

Insert the number of participants and the main characteristics of them

RESPONSE: YOUR SUGGESTION HAS BEEN FOLLOWED.

Introduction

The introduction is quite long and some aspects should be reviewed. For example in the 3rd paragraph, none aspect regarding the football game is considered on training plain. In any momento of the introduction authors retract this important situation that is included in football players, the effort during the games. We know that the training of this athletes is based on the calendar of the games. Please, rewrite some parts of the introduction considering the above mentioned aspect.

RESPONSE: DEAR REVIEWER, THANK YOU. WE HAVE TRIED TO TURN SMALLER THE PARAGRAPH AND MERGE WITH THE NEXT ONE.  

In the last paragraph, authors should describe the principal variables that were included in PCA analysis in the others studies and justify the use of them.

 RESPONSE: DEAR REVIEWER, THANK YOU. WE HAVE ADDED EXAMPLES OF THE MAIN OUTCOMES IN THE BOTTOM OF THE OBJECTIVES.

Methods

Authors should explain how was the PCA method? Which variables were included and excluded? Authors should explain step by step.

 RESPONSE: YOUR SUGGESTION HAS BEEN FOLLOWED IN THE STATISTICAL ANALYSIS SECTION. THE TWO ARTICLES WHERE THE METHODDOLOGY WAS DETAILED STEP BY STEP WERE INCLUDED FOR MORE DETAILS.

Participants

Authors should explain how was the control of the games during the evaluation period.

 RESPONSE: YOUR SUGGESTION HAS BEEN FOLLOWED. IF ANY OTHER ISSUE IS SUGGESTED, PLEASE LET US KNOW.

Protocol

Authors should better explain the experimental protocol, focus on the efforts in different training tasks. A lot of information was described about the instruments, but a few information about the tasks proposed to evaluate the efforts.

 RESPONSE: YOUR SUGGESTION HAS BEEN FOLLOWED. ALL TABLES RELATED TO TASK HAVE BEEN ADDED TO THIS SECTION. THE REMAINING INFORMATION WAS DETAILED IN THIS PARAGRAPH.

A figure regarding the experimental protocol and the methods used should be included.

 RESPONSE: THAN YOU VERY MUCH. A FIGURE WAS INCLUDED.

Measurements 

Authors should include the error of measurements using IMU.

RESPONSE: YOUR SUGGESTION HAS BEEN FOLLOWED.

Discussion

See the comments in the introduction section to improve the discussion.

RESPONSE: THANK YOU FOR THIS SUGGESTION. WE CHECKED THE DISCUSSION SECTION ACCORDING TO YOUR COMMENTS.

Reviewer 2 Report

Congratulations on the work done.

This manuscript is well structured and organized.

The methodology is clear and the data analysis used is adequate to the objectives of the work. The results obtained can help teams and training preparation.

Author Response

Congratulations on the work done.

This manuscript is well structured and organized.

The methodology is clear and the data analysis used is adequate to the objectives of the work. The results obtained can help teams and training preparation.

RESPONSE: THANK YOU VERY MUCH FOR YOUR ACKNOWLEDGEMENT AND SUPPORTING OUR WORK.

Reviewer 3 Report

The manuscript presented for review is very well prepared methodologically. The data collection is very thorough and the distribution of indicators is logical. This is the first work so widely analyzing the external load in futsal using LPS technology. The topic of the work, tables are correct and reflect the content of the work. From the duty of a reviewer I would like to draw attention to the issues that require supplementation or editorial ordering:

- the limitation section should be separated and the content of the discussion should be placed there

- It should be justified that such a small sample as 14 players authorizes to conduct such an analysis of loads. There is no further justification for the choice of statistical methods for the purposes stated.

- Internal load is analyzed on the basis of % HRmax. No sample, which would determine HRmax for each of the subjects. 

Despite these remarks the work is very innovative and should be evaluated positively.

Author Response

The manuscript presented for review is very well prepared methodologically. The data collection is very thorough and the distribution of indicators is logical. This is the first work so widely analyzing the external load in futsal using LPS technology. The topic of the work, tables are correct and reflect the content of the work. From the duty of a reviewer I would like to draw attention to the issues that require supplementation or editorial ordering:

RESPONSE: THANK YOU VERY MUCH FOR YOUR ACKNOWLEDGEMENT AND SUPPORTING OUR WORK.

- the limitation section should be separated and the content of the discussion should be placed there

RESPONSE: YOUR SUGGESTION HAS BEEN FOLLOWED.

- It should be justified that such a small sample as 14 players authorizes to conduct such an analysis of loads. There is no further justification for the choice of statistical methods for the purposes stated.

RESPONSE: THANK YOU VERY MUCH FOR HIGHLIGHTING THIS ISSUE. IT WAS JUSTIFIED IN THE PARTICIPANT SECTION: “This sample was considered due to the high number of individual observations and the real-world scientific practice in high-level settings”

- Internal load is analyzed on the basis of % HRmax. No sample, which would determine HRmax for each of the subjects. Despite these remarks the work is very innovative and should be evaluated positively.

RESPONSE: THANK YOU VERY MUCH FOR YOUR ACKNOWLEDGEMENT AND SUPPORTING OUR WORK.

Round 2

Reviewer 1 Report

The paper entitled "The most demanding efforts in different training tasks in professional female futsal: A mid-season study through principal component analysis" now is well written and objetive. The authors improved a lot all sections indicated for correction. 
